# Effects of Surface Modification on Adsorption Behavior of Cell and Protein on Titanium Surface by Using Quartz Crystal Microbalance System

**DOI:** 10.3390/ma14010097

**Published:** 2020-12-28

**Authors:** Takumi Matsumoto, Yuichiro Tashiro, Satoshi Komasa, Akiko Miyake, Yutaka Komasa, Joji Okazaki

**Affiliations:** 1Department of Removable Prosthodontics and Occlusion, Osaka Dental University, 8-1 Kuzuha-hanazono-cho, Hirakata, Osaka 573-1121, Japan; matsumoto-t@cc.osaka-dent.ac.jp (T.M.); tashiro@cc.osaka-dent.ac.jp (Y.T.); joji@cc.osaka-dent.ac.jp (J.O.); 2Department of Japan Faculty of Health Sciences, Osaka Dental University, 1-4-4, Makino-honmachi, Hirakata-shi, Osaka 573-1121, Japan; miyake-a@cc.osaka-dent.ac.jp (A.M.); komasa@cc.osaka-dent.ac.jp (Y.K.)

**Keywords:** titanium, UV-treatment, atmospheric pressure plasma-treatment, QCM measurement, ROS assay

## Abstract

Primary stability and osseointegration are major challenges in dental implant treatments, where the material surface properties and wettability are critical in the early formation of hard tissue around the implant. In this study, a quartz crystal microbalance (QCM) was used to measure the nanogram level amount of protein and bone marrow cells adhered to the surfaces of titanium (Ti) surface in real time. The effects of ultraviolet (UV) and atmospheric-pressure plasma treatment to impart surface hydrophilicity to the implant surface were evaluated. The surface treatment methods resulted in a marked decrease in the surface carbon (C) content and increase in the oxygen (O) content, along with super hydrophilicity. The results of QCM measurements showed that adhesion of both adhesive proteins and bone marrow cells was enhanced after surface treatment. Although both methods produced implants with good osseointegration behavior and less reactive oxidative species, the samples treated with atmospheric pressure plasma showed the best overall performance and are recommended for clinical use. It was verified that QCM is an effective method for analyzing the initial adhesion process on dental implants.

## 1. Introduction

Various materials are currently used for dental implants, such as metals and ceramics, although Ti and its alloys are most commonly used, as they have low density, excellent biocompatibility, corrosion resistance, and appropriate mechanical properties [1]. Fast osseointegration of the implant is critical for successful clinical outcomes and depends on both the body conditions and surface morphology and properties of the material. It is desirable to shorten the healing period because it can take several months to half a year or more to achieve osseointegration after implantation. Therefore, many studies have tried to accelerate osseointegration by surface-treating Ti materials [2,3,4,5]. Komasa et al. reported that the surface treatment of Ti and zirconia implant materials by ultraviolet (UV), atmospheric-pressure plasma, and alkali treatments can promote the early formation of hard tissue in the tissue surrounding the implant [6,7]. Other commonly used surface-treatment methods include acid etching [8,9], sandblasting [10], anodization [11], physical vapor deposition [12], calcium phosphate coating [13,14,15], and hydroapatite coating [16,17,18,19].

Specifically, the surface properties and wettability of the implant surface are involved in the early formation of hard tissue around the implant [20,21,22,23]. For example, the hydrophilization of Ti implants promotes osseointegration [24,25], soft tissue adhesion [26,27], cell adhesion, and bone protein adhesion [28,29] and suppresses inflammatory cytokine aggregation [30,31] and biofilm formation [32]. Methods for hydrophilizing the Ti surface include physical methods such as UV-treatment [33,34,35,36] and low temperature plasma treatment (plasma, glow discharge) [37,38,39], and chemical methods such as blasting + acid etching [40,41], hydrogen peroxide (H_2_O_2_) solution treatment [42,43], and sodium hydroxide (NaOH) solution treatment [44]. UV and plasma treatments are advantageous for application in clinical practice due to the low risk of contamination. In the low temperature plasma method, free molecules are dissociated in a weak electric field under low pressure to generate plasma. For example, under atmospheric pressure plasma processing generates ionizing electrons and cations by applying a high voltage to air. Ozone is generated from the plasma and the redox reaction by active oxygen generated from the plasma causes the decomposition of organic matter and generates hydroxyl groups [45,46]. In the UV irradiation method, oxygen radicals generated by photo catalysis immediately oxidize at the site of the molecular bond in the organic compound cleaved by the UV radiation to generate a hydroxyl group [47]. These reactions result in surface activation, which increases the adhesiveness and wettability of the surface. However, as few studies have compared these methods in detail, clinicians have limited information for selecting the most appropriate treatment. Previous studies have proposed different opinions regarding the effects of both bone marrow cells on the ability to induce hard tissue differentiation. Therefore, it is necessary to examine the early fixation behavior to confirm the differences between the two methods.

A quartz crystal microbalance (QCM) can be used for measuring the initial adsorption behavior of cells and proteins as it can quantify the amount of adsorption on the sensor surface at the nanogram level, detect the interaction of biomolecules such as proteins, and characterize the adsorption reaction in real time. Studies using QCM measurements are common in the engineering field, but there are few reports of its use in dentistry. Miyake et al. applies QCM to dentistry research and has developed a QCM sensor that mimics denture materials and quantifies the amount of protein adsorbed on the sensor surface [48]. It was previously reported that QCM is effective for quantifying the adsorption behavior of proteins and bone marrow mesenchymal cells on surface treated Ti implants [49].

In this study, the initial behavior of bone marrow cells and proteins on the surface of Ti materials treated by UV and atmospheric pressure plasma treatments was compared, to clarify the differences between these methods. The efficacy of such treatment methods was verified and it was concluded that atmospheric pressure plasma treatment provided the best surface properties for promoting early fixation and osseointegration.

## 2. Materials and Methods

### 2.1. Study Design

ALP activity and calcium deposition of rat bone marrow (RBM) cells on Ti disks subjected to different surface treatments was evaluated. In order to investigate the reason for this difference in biocompatibility, the initial behavior of the cells and proteins was focused on immediately after implantation. By using the QCM system, the adsorption behavior of proteins and RBM cells on Ti-QCM sensors with different surface treatments was compared. In order to examine the results of the QCM analysis, surface analysis and ROS analysis of the material surface were performed, and the events that occurred were considered.

### 2.2. Sputtering Procedure

To prepare the Ti-QCM sensors (simulating implant materials), the bare QCM sensors were first cleaned using a piranha solution (sulfuric acid: 30% hydrogen peroxide = 7:3 *v*/*v*) and ultrasonically cleaned with high purity acetone (99.999%). To prepare the target, pure Ti powder was pressed into a 75 mm diameter disk. A thin layer of Ti was deposited on a quartz disk (diameter: 8 mm, area: 4.9 mm^2^) by reactive magnetron sputtering using a radio-frequency magnetron sputtering system (CFS-4ES-231; Shibaura Mechatronics Co., Ltd., Kanagawa, Japan). The bare QCM sensors were positioned 85 mm above the target and the magnetron sputtering chamber was evacuated to a pressure of 3 × 10^−3^ Pa. Argon was used for sputtering and its pressure was maintained at 6.7 × 10^−1^ Pa. The Ti target was discharged at 480 W, resulting in a Ti layer with a thickness of approximately 240 nm. The Ti-QCM sensors were cleaned with sodium dodecyl sulfate and UV-ozone cleaner (PC450; Meiwafosis Co., Ltd., Osaka, Japan) before the QCM measurements.

### 2.3. Sample Preparation

Three types of Ti-QCM sensors were prepared: bare, UV-treated, and plasma-treated. In the UV-treated group, Ti-QCM sensors and disks (Daido Steel, Osaka, Japan) were irradiated with UV light (wavelength = 254 nm, power = 100 mW/cm^2^) for 15 min using a UV irradiation system (HL-2000 HybriLinker; Funakoshi, Tokyo, Japan). The plasma-treated Ti-QCM sensors were irradiated with low temperature plasma (Piezobrush^®®^ PZ2; Relyon Plazma GmbH, Regensburg, Germany) under irradiation at 0.2 MPa for 30 s at 10 mm.

### 2.4. Sample Characterization

Scanning electron microscopy (SEM; S-4800, Shimadzu, Kyoto, Japan) and scanning probe microscopy (SPM; SPM-9600, Shimadzu, Kyoto, Japan) were used to analyze the surface topology and roughness of the Ti-QCM sensors over an area of interest of 2.0 × 2.0 μm^2^. X-ray photoelectron spectroscopy (XPS; ESCA 5600, Ulvac-Phi Inc., Kanagawa, Japan) was used to analyze the chemical composition of the surface of the Ti-QCM sensors. AlKα radiation (15 kV, 300 W) was used as the X-ray source. During XPS, Ar ion sputtering was used to determine the composition of the surface layer.

Contact angle measurements of the Ti-QCM sensors were performed using a video contact angle measurement system (SImage Entry 6; Excimer Inc., Kanagawa, Japan). The measurement was performed after 2.6 μL of distilled water was dropped immediately after the surface treatment on the Ti-QCM sensors. The contact angle was used as a simple measure of the surface energy and hydrophilic nature of the sensor surfaces.

### 2.5. Cell Culture

Sprague-Dawley rats (8 weeks old) were used to isolate rat bone marrow cells (RBMCs). The rats were euthanized using 4% isoflurane. The bones were aseptically extracted from the hind limbs. The proximal end of the femur and distal end of the tibiae were removed. The bones were rinsed with Eagle’s minimal essential medium (MEM; Wako Pure Chemical Industries, Ltd., Osaka, Japan). Then, the RBMCs were aspirated using a 21-gauge needle (Terumo, Tokyo, Japan). The bone marrow pellets were separated by trituration and cell suspensions from all bones were combined by centrifugation. The RBMCs were cultured in 75 cm^2^ Falcon culture flasks (BD Biosciences; Franklin Lakes, NJ, USA) at 37 °C and 5% CO_2_ atmosphere. Growth medium containing MEM was supplemented with 10% fetal bovine serum (FBS; Invitrogen/Life Technologies, Carlsbad, CA, USA), penicillin (500 U mL^−1^; Cambrex Bio Science, Walkersville, MD, USA), and Fungizone (1.25 μg mL^−1^; Cambrex Bio Science, Walkersville, MD, USA). At confluence, trypsin was used to remove the cells from the flask, followed by washing with phosphate-buffered saline (PBS) twice. The cells were resuspended in a culture medium at a concentration of 4 × 10^4^ cells/cm^2^ in a 24-well tissue culture plate (BD Biosciences, Franklin Lakes, NJ, USA). This study was approved by the Guidelines for Animal Experimentation of Osaka Dental University (Approval No. 20-08001).

### 2.6. QCM Measurements

In QCM, a crystal oscillator vibrates at a certain frequency (resonant frequency) when an AC electric field is applied to a thin plate-shaped crystal with metal electrodes attached to both sides. The frequency of this crystal plate changes according to the mass of the substance on the electrode. The frequency decreases as the amount of attached material increases, and vice versa. Using this phenomenon, changes in the QCM frequency (Δ*F*) are measured to determine the mass change of the substance on the electrode (Δ*m*). In QCM, Δ*F* depends on the adsorbed mass following Sauerbrey’s equation [50]:(1)$$\Delta F=-\frac{2F_{0}^{2}}{\sqrt{\rho_{q}\mu_{q}}}\frac{\Delta m}{A}$$
where, *F*_0_ is the fundamental frequency of the crystal (27 × 10^6^ Hz), *A* is the electrode area (0.049 cm^2^), *ρ_q_* is the quartz density (2.65 g cm^−3^), and *μ_q_* is the shear modulus of quartz (2.95 × 10^11^ dyn cm^−2^). This equation shows that a Δ*F* of 1 Hz at 27 MHz is associated with a mass difference of 0.62 ng cm^−2^.

The initial behavior of the implant surface was evaluated using the QCM method with two proteins involved in osseointegration, namely bovine serum albumin (BSA; Wako Pure Chemical Industries, Ltd., Osaka, Japan) and human plasma fibronectin (HFN; Nakarai Tesque, Inc., Kyoto, Japan). The BSA and HFN were dissolved separately in PBS (pH 7.4) to prepare 200 or 500 μg/mL solutions, respectively. Quantification of the amount of adsorbed proteins (BSA and HFN) and RBM cells was performed using QCM measurement (Affinix QNμ; Initium Co., Ltd., Tokyo, Japan). This system has 550 µL cells, where the bottom of the cell was equipped with a 27 MHz QCM plate. In the QCM sensors, the crystal plate had a diameter of 8 mm and the area of the Au-plated crystal was 4.9 mm^2^. This device was equipped with a stirrer bar and a temperature controller, and the frequency change was measured with a universal frequency counter attached to the microcomputer. Each Ti-QCM sensor was immersed in 500 μL of PBS solution (0.01 mol/L PBS at pH 7.4). The changes in the QCM frequency (Δ*F*) were recorded as a function of time, where the measurement was started immediately after the injection of 5 μL (20 μg/mL) of BSA, HFN, or RBM cells. The protein diffusion effect did not affect the measurements of the stirred solution. Hence, the stirring did not affect the frequency stability or the magnitude of Δ*F*. The QCM measurements were performed at 25 °C.

### 2.7. Cell Morphology

RBMCs were seeded on Ti-QCM sensors. The QCM sensors with attached cells were washed with PBS and fixed with 4% paraformaldehyde for 20 min at room temperature. Proteins were lysed with 0.2% Triton X-100 for 30 min and 6 h at room temperature. Blocking one reagent (Nacalai Tesque, Kyoto, Japan) was added to the cells, followed by incubation at room temperature for 30 min. Alexa Fluor 488-Phalloidin (Invitrogen, Life Technologies, Carlsbad, CA, USA) and DAPI were used to stain treated cells at 37 °C in the dark for 1 h. F-actin and the cell nuclei were visualized using an LSM 700 confocal laser scanning microscope (Carl Zeiss Ltd., Oberkochen, Germany).

### 2.8. ALP Activity

Alkaline phosphatase (ALP) activity was analyzed using RBSCs seeded on Ti disks and cultured for 7 and 14 days. The cultured cells were washed with PBS. The protein was dissolved by adding 300 μL of 0.2% Triton X-100. The mixture was stirred for 20 s at 29 Hz using a shaker (Mixer Mill Type MM 301; Retsch GmbH, Hahn, Germany) to determine the lysate content. An alkaline phosphatase luminescence enzyme-linked immunosorbent assay (ELISA) kit (Sigma-Aldrich, St. Louis, MO, USA) was used to measure ALP activity. The experiment was performed according to the manufacturer’s protocol. To terminate the reaction, 50 μL of 3 N NaOH was added. Quantification of p-nitrophenol release was determined by measuring the absorbance at 405 nm using Spectra Max^®®^ M5 (Molecular Devices Ltd., San Jose, CA, USA). A PicoGreen dsDNA assay kit (Invitrogen/Life Technologies, Carlsbad, CA, USA) was used to measure DNA content. The experiment was performed according to the manufacturer’s protocol. The ALP activity was normalized to the DNA content in the cells.

### 2.9. Mineralization

RBMCs were seeded on Ti disks for calcium determination and cultured for 21 and 28 days. An E test kit (Wako Pure Chemical Industries, Ltd., Osaka, Japan) was used for the analysis. To 50 µL of medium, 1 mL of calcium E-test reagent and 2 mL of buffer were added. A 96 microplate reader (SpectraMax M5; Molecular Devices, Sunnyvale, CA, USA) with a wavelength of 610 nm was used to measure the absorbance of the reaction products. The calcium ion concentration was calculated from the absorbance values for the standard curve.

### 2.10. ROS Assay

An experiment was conducted to analyze the reactive oxygen species (ROS). RBMCs were seeded on Ti disks and incubated for 3 days. The medium was removed and immersed in 1 mL of medium containing 5 µM Cell ROX^®®^ oxidative stress reagent (C10422, Thermo Fisher Life Technologies Limited, Tokyo, Japan). The samples were incubated at 37 °C for 15 min and then washed 3 times with PBS and fixed with 4% paraformaldehyde at room temperature for 15 min. The RBMCs were viewed using an LSM700 confocal laser scanning microscope (Carl Zeiss Ltd., Oberkochen, Germany). The software provided with the confocal laser scanning microscope was used to analyze the fluorescence intensity of the observed images.

### 2.11. Statistical Analysis

Each measurement was performed three times and statistical analyses were performed by one-way analysis of variance. When a significant difference was found, Bonferroni’s multiple comparison was used. The significance level was <5%.

## 3. Results

### 3.1. Surface Structure

The SEM images in Figure 1 show that no significant changes in the surface microstructure of Ti-QCM sensors were observed after UV or plasma treatment. Similarly, the SPM images in Figure 2 show that the surface roughness of the samples was similar, with surface roughness (*Ra*) values of the bare, UV-treated, and plasma-treated Ti-QCM sensors of 4.708, 4.696, and 3.890 nm, respectively.

### 3.2. Chemical Composition

Figure 3 shows wide-scan XPS results (elemental composition) of the surfaces of the bare, UV-treated, and plasma-treated Ti-QCM sensors. The presence of Ti, O, and C were confirmed on the surface of all Ti-QCM sensors. The UV-treated and plasma-treated sensors showed higher O contents and lower C content than the untreated sensor, where the plasma-treated sample had the highest O content.

### 3.3. Surface Tension

Figure 4 shows the cross-sectional images of water droplets on the three different contact angle of the bare, UV-treated, and plasma-treated Ti-QCM sensors of 90.6°, 7.2°, and 0°, respectively. The bare sample was quite hydrophobic, while the surface of the UV-treated Ti was hydrophilic, and that of the plasma-treated Ti was super-hydrophilic (contact angle < 5°).

### 3.4. Cell Adsorption

Figure 5 shows the measured adsorbed weight of albumin, fibronectin, or RBM cells on each sensor type for 60 min following the injection. The adsorbed weight increased rapidly over the first 30 min, followed by a slower increase thereafter up to and beyond 60 min. Similar to the total adsorbed weights, the adsorption was the highest for the plasma-treated sensor over the entire measurement time for all cell types, followed by the UV-treated and untreated samples.

Figure 6 shows the amounts of adsorbed albumin, fibronectin, and RBM cells determined using QCM measurements. Immediately after the injection of the cells, a decrease in the frequency of the QCM system was observed, indicating the adsorption of the cells on the Ti-QCM sensors. The frequency stabilized over 60 min. For all cell types, the highest total adsorption was measured for the plasma-treated Ti-QCM sensor, followed by the UV-treated and untreated sensors.

### 3.5. Cell Morphology and ALP Activity

Figure 7 shows the cell morphology observed by phalloidin and DAPI staining after 24 h incubation. Greater F-actin expression and more filopodia and lamellipodia were observed in RBM cells on the UV-treated and plasma-treated Ti surfaces than on the untreated Ti.

Figure 8 shows the ALP activity of RBM cells seeded on untreated, UV-treated, and plasma-treated Ti disks after 7 and 14 days. Among the three samples, the ALP activity was highest for the plasma-treated sample and lowest for the untreated sample. The ALP activity of the RBM cells was higher on the 14th day than the 7th day.

### 3.6. Mineralization

Figure 9 shows the Ca deposition of RBM cells seeded on untreated, UV-treated, and plasma-treated Ti disks after 21 and 28 days. Among the three samples, Ca deposition was the highest for the plasma-treated and lowest for the untreated samples. Mineralization of RBM cells on day 28 was higher than on day 21.

### 3.7. ROS Assay

Figure 10 shows the levels of intracellular ROS for cells seeded on untreated, UV-treated, and plasma-treated Ti disks. The greatest ROS accumulation was observed for the untreated sample, while the lowest accumulation was observed for the plasma-treated sample.

## 4. Discussion

To evaluate the effectiveness of various surface modification methods for hydrophilizing implant surfaces, simulated the initial hard-tissue-formation behavior of the tissue surrounding the implant material. SPM and SEM results showed that there was no difference in the surface morphology between the untreated and treated samples, indicating that XPS results showed a decrease in C content and an increase in O and after the two treatments, while the plasma-treated sample had higher O than the UV-treated sample. The contact angle measurements showed that surface hydrophilicity was achieved after the UV and plasma treatments because of reduction of carbon. The plasma-treated Ti surface showed the lowest contact angle, which enabled the highest adsorption of albumin, fibronectin, and RBM cells. While the performance of the UV-treated and plasma-treated samples were generally similar, the plasma-treated samples showed the highest amounts of F-actin, filopodia, and lamellipodia, the highest ALP activity, the highest amount of Ca precipitation, and the lowest ROS value of all sample types.

These findings are consistent with those of various previous studies, which report that the hydrophilization of the surface of Ti implants greatly enhances the initial adhesion/proliferation and differentiation-inducing ability of osteoblasts. The hydrophilic surface facilitates blood flow over the surface and enhances the initial adhesion of bone marrow cells. As the degree of contamination and hydrophilicity of the Ti surface affect the bone bond strength of the implant, minimizing the degree of contamination and imparting hydrophilicity are important directions for future implant development. UV and plasma treatments impart hydrophilicity to the Ti surfaces while removing contaminants. Various studies [51,52] have demonstrated higher initial adhesion and hard tissue differentiation on treated Ti surfaces compared to untreated ones. Similarly, it was observed that both treatment processes resulted in enhanced ALP activity and calcium precipitation compared to the untreated Ti surface, where the plasma-treated samples showed the best performance. Our previous reports [53,54,55] showed that both methods result in Ti surfaces with high neonatal bone formation potential in vivo.

The first step in the osseointegration process is the blood-mediated adsorption of proteins on the Ti implants, followed by osteoblast accumulation and differentiation. The hard tissue surrounding the implant is formed while expressing various differentiation markers [56]. Therefore, the observation of the initial behavior of protein and cell adsorption on Ti implants is important for studying osseointegration. QCM was chosen to measure trace amounts of protein and cells immediately after their adsorption on the Ti implant as it was previously shown to be an effective method [57]. Tashiro et al. performed surface modification of implant materials and analyzed protein and cell adsorption by QCM measurements [49].

In this study, the surface-modified Ti samples showed significantly more protein and cell mass than the untreated ones, with the plasma-treated Ti showing the best adsorption behavior. The hydrophobic surface was attributed to the surface treatment methods removing adventitious C content (which increases the surface energy). Furthermore, the plasma treatment resulted in a higher surface O content than the UV-treatment, which increased the hydrogen bonds involved in the hydroxyl groups, resulting in an increase in the amount of protein and cells adsorbed. It was observed that the proteins and cells began to adsorb on the Ti surface immediately, and rapid adsorption was observed for the first 30 min, followed by slower adsorption over the next 30 min. This suggests that the Ti surface changed from hydrophilic to hydrophobic immediately after UV and plasma treatments.

In this study, it was observed that the morphology of the adsorbed RBM cells on the Ti surfaces. Many studies [58,59] have confirmed the adsorption behavior of proteins and cells using QCM measurements, but there are few studies that confirm whether cells are adsorbed after confirming the adsorption behavior. Surface treated Ti showed significantly higher F-actin expression and more filamentous and foliate pseudopodia than untreated surfaces, where the plasma-treated Ti showed the highest concentrations. Wang et al. performed plasma treatment and Sugita et al. reported that UV treatment increases the adhesion strength of cells [57,58]. The Ti-QCM sensors were extremely thin and had a small area. Nevertheless, strong adhesion of cells was observed because it is considered that the adhesion strength of cells was increased by imparting hydrophilicity, as described above. Our QCM results showed that UV and plasma treatment were both effective for increasing the adsorption of proteins and the initial adhesion of cells on Ti implants for enhanced osseointegration, where the plasma treatment showed the best overall performance.

It is considered that the difference in the behavior of cells and proteins on the surface of the Ti samples was related to surface changes. In general, the surface of implant materials can be modified via changes to the surface texture or shape. The surface treatment methods used here do not impart major changes to the surface shape (as confirmed by SEM and SPM analyses) and insignificant changes in the surface roughness were observed. XPS analysis showed that both treatments reduced the C content (i.e., surface contaminants) and increased the O, with the largest effect for the plasma treatment. A decrease in C on the surface of the material is said to create an environment that improves bone marrow cell adhesion and proliferation. In addition, it has been reported that the formation of an oxide film showing an increase in O is involved in the induction of bone marrow cell differentiation. In both sides, the results of this experiment show that both treatments are effective for osseointegration. In the evaluation of surface wettability, it was clarified that both treatments produced highly hydrophilic surfaces, with super hydrophilicity achieved by the plasma-treated samples. The generated active oxygen species decompose the hydrocarbons adsorbed on the surface of the material and react with the hydrogen of the hydrocarbon molecules and water molecules in the atmosphere adsorbed on the surface to form surface hydroxyl groups, which results in a super hydrophilic surface of titanium oxide. In addition, it has been reported that a super hydrophilic surface is formed in an extremely short time because the atmospheric-pressure plasma treatment has a very high energy compared to the UV-treatment [60,61], and the many studies obtained in this experimental result is considered to be involved in the difference between barometric plasma treatment and UV treatment. Many previous reports have clarified the relationship between the hydrophilicity of the material surface and the initial adhesion of bone marrow cells and the ability to induce hard tissue differentiation [62]. Hydrophilic surfaces promote adsorption of fibronectin, which is an adhesive protein, which in turn promotes cell adhesion, followed by cell protrusion elongation, and finally the implant, which results in bone contact with implant material.

Implant treatments always involve surgical procedures, where oxidative stress from inflammation can affect the surrounding tissue. While ROS is important for in vivo remodeling, excessive amounts are known to cause delayed cell apoptosis and tissue healing [63], and should be controlled to acceptable levels for successful implant treatment. Some studies have reported that implant surface treatments suppress ROS generation by generating active oxygen [64]. In this study, evaluation of ROS using bone marrow cells revealed that both treatment methods suppressed the generation of ROS on the Ti surface, although the plasma-treated material showed the largest effect. Another report showed that the suppression of ROS production on the surface of the material is caused by the reduction of C [65]. The experimental ROS results correlated well with the XPS elemental analysis results.

It is clear that both UV-treatment and atmospheric pressure plasma treatment are useful methods for imparting hydrophilicity to the implant surface and enhancing initial fixation after implant placement. Although the results of this study revealed there was a difference in the effect of each treatment. It was not possible to explain the reason why the amount of adhesion increased rapidly 5 min after the appropriate amount. Further consideration is needed on the reason for these results. Although both treatments showed similar results, atmospheric pressure plasma treatment may be a better choice for clinical dentists. The piezobrush used to apply the plasma-treatment is more compact than the conventional UV irradiation device, and has a higher energy, which could provide advantages for initial fixation after implant placement. It was verified that the QCM system was suitable for analyzing the attachment mechanism of bone marrow cells and proteins immediately after implantation. It was also hoped that the high energy of the treatment will make it clear that atmospheric pressure plasma treatment is superior, which will be useful information for clinical dentists.

## Figures and Tables

**Figure 1 materials-14-00097-f001:**
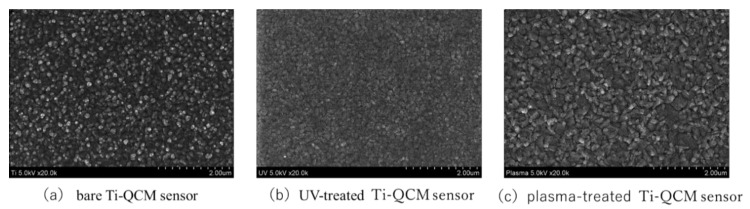
SEM images of (**a**) bare, (**b**) UV-treated, and (**c**) plasma-treated Ti quartz crystal microbalance (QCM) sensors.

**Figure 2 materials-14-00097-f002:**
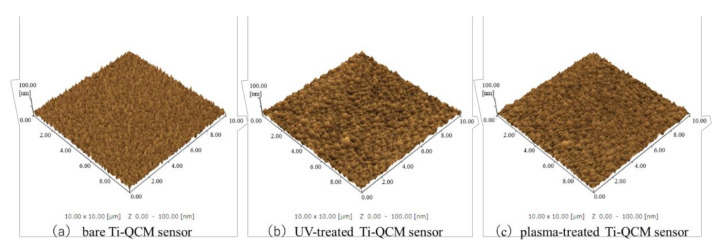
SPM images of (**a**) bare, (**b**) UV-treated, and (**c**) plasma-treated Ti-QCM sensors.

**Figure 3 materials-14-00097-f003:**
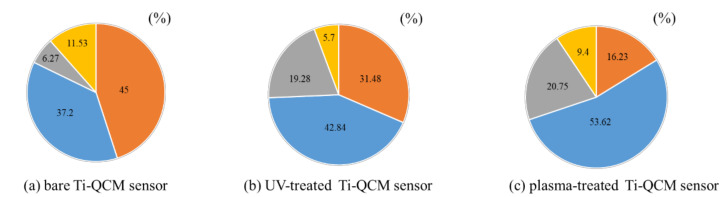
Results of XPS elemental analysis of (**a**) untreated, (**b**) UV-treated, and (**c**) plasma-treated Ti-QCM sensors. Orange = C1s, blue = O1s, gray = Ti2p, and yellow = other elements.

**Figure 4 materials-14-00097-f004:**

Photographs of the contact angles of water on the surfaces of the (**a**) bare, (**b**) UV-treated, and (**c**) plasma-treated Ti-QCM sensors.

**Figure 5 materials-14-00097-f005:**
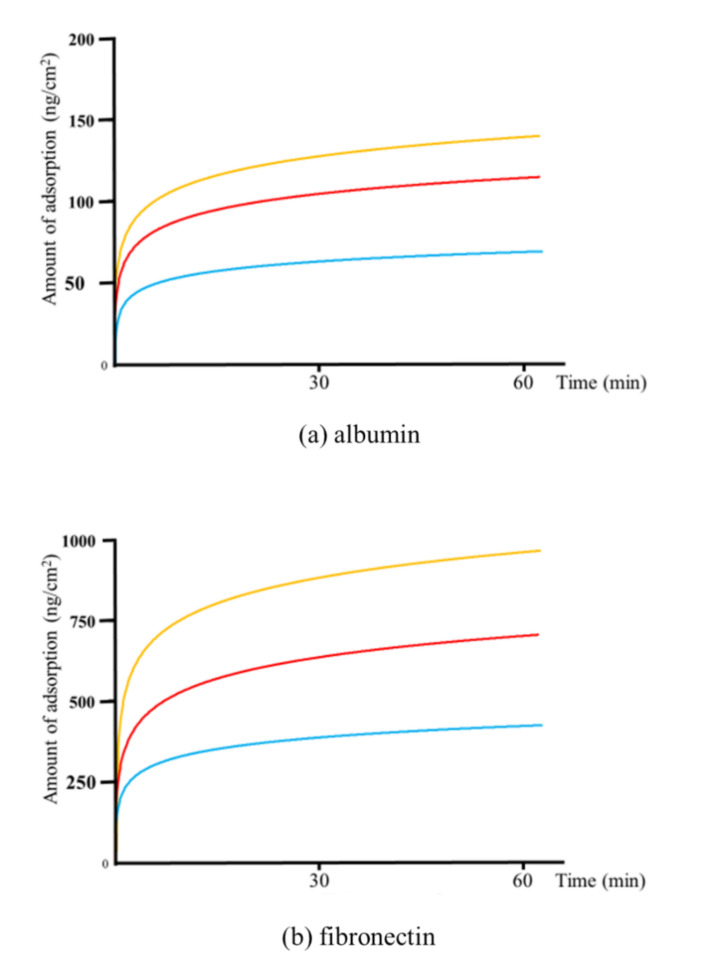
Changes in the adsorption over time of (**a**) albumin, (**b**) fibronectin, and (**c**) rat bone marrow (RBM) cells on (blue) untreated, (red) UV-treated, and (yellow) plasma-treated Ti-QCM sensors.

**Figure 6 materials-14-00097-f006:**
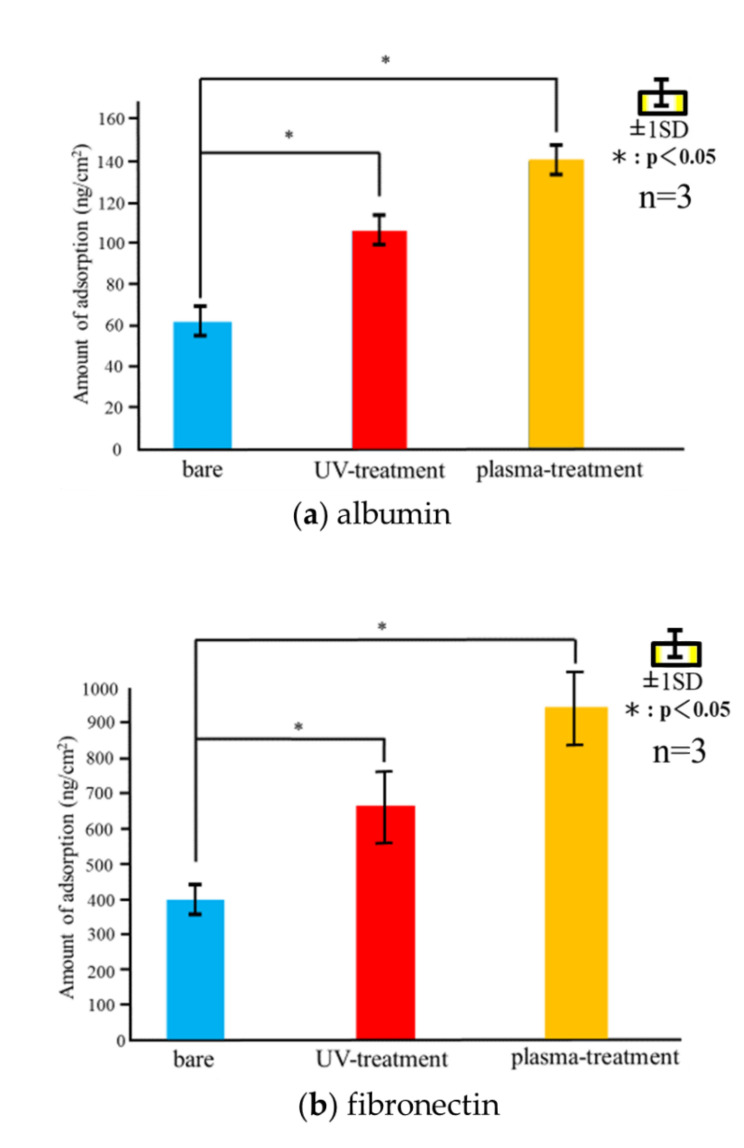
Final adsorption weights of (**a**) albumin, (**b**) fibronectin, and (**c**) RBM cells on the three sample types after 60 min adsorption.

**Figure 7 materials-14-00097-f007:**
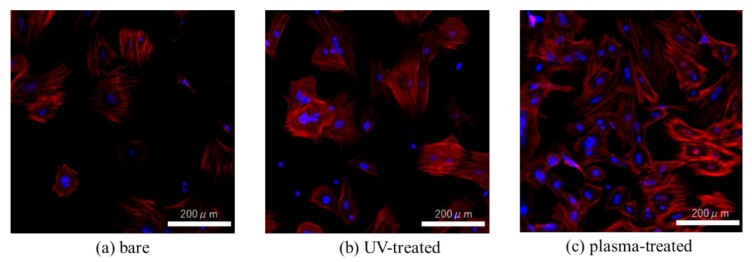
Fluorescent stained image of (**a**) bare, (**b**) UV-treated, (**c**) plasma-treated Ti-QCM sensors were seeded with rat bone marrow mesenchymal stem cells and stained with phalloidin (F-actin) and DAPI (nucleus) 24 h later.

**Figure 8 materials-14-00097-f008:**
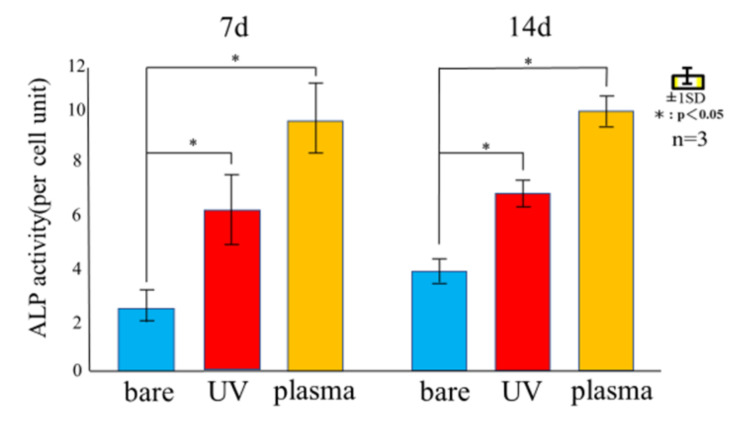
ALP expression in RBM cells after 7 and 14 days of culturing on untreated, UV-treated, and plasma-treated Ti disks.

**Figure 9 materials-14-00097-f009:**
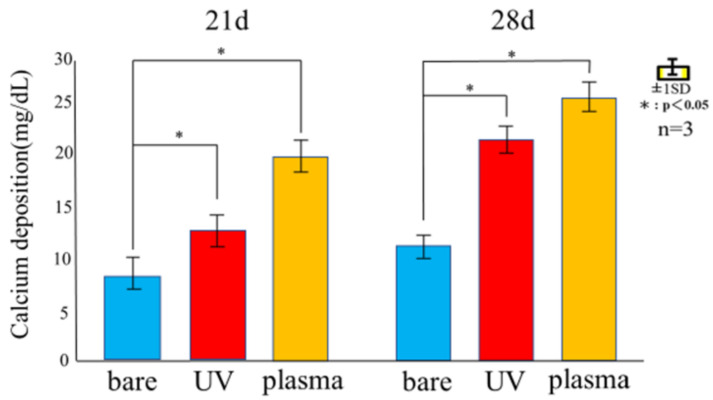
Amount of calcium deposited 21and 28 days after incubation of the culture on untreated, UV-treated, and plasma-treated Ti disks.

**Figure 10 materials-14-00097-f010:**
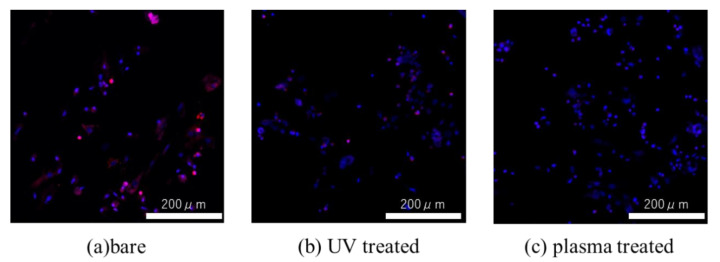
Fluorescent stained images of intracellular ROS of RBM mesenchymal stem cells seeded on (**a**) untreated, (**b**) UV treated, and (**c**) plasma-treated Ti disks.

## Data Availability

Data sharing is not applicable to this article.

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
