# Peer review of "Effects of Surface Modification on Adsorption Behavior of Cell and Protein on Titanium Surface by Using Quartz Crystal Microbalance System"

_materials, 2020, doi:10.3390/ma14010097_

Round 1
Reviewer 1 Report
Dear authors, thanks to provide a well conducted research. In order to improve the readability of your manuscript, I suggest minor revision, making the research easy to read for both researcher and clinicians.
Title
- Explain QCM in the title and report the study design.
Abstract
- Early fixation. What does it mean? Primary stability? Please clarify.
- Report the study design in the title. Authors reported "used to measure the nanogram-level amount of protein and bone marrow cells adhered to the surfaces of Ti implants in real time." Later, authors reported "...QCM sensors (simulating implant materials)" I failed to understand the study design.
- "surface C content and increase in the O content". Fedine C and O contents.
- "various analyses showed". Which analyses?
- "We verified..." avoid personal pronoun.
Introduction
- "Our group..." Avoid personal pronoun
- Too many references from in vitro studies. Please, also report references from in vivo researches, such as:
Park C-J, Lim JH, Tallarico M, Hwang K-G, Choi H, Cho G-J, et al. Coating of a Sand-Blasted and Acid-Etched Implant Surface with a pH-Buffering Agent after Vacuum-UV Photofunctionalization. Coatings. 2020;10(11):1040.
Tallarico M, Baldini N, Gatti F, Martinolli M, Erta X, Meloni SM, et al. Role of New Hydrophilic Surfaces on Early Success Rate and Implant Stability: 1-Year Post-loading Results of a Multicenter, Split-Mouth, Randomized Controlled Trial. European Journal of Dentistry. 2020 Jul 31;1–10.
- "In this study, we compared the initial behavior of bone marrow cells and proteins on the surface of titanium materials treated by UV and atmospheric-pressure plasma treatments, to clarify the differences between these methods. We verified the efficacy of such treatment methods and concluded that atmospheric-pressure plasma treatment provided the best surface properties for promoting early fixation and osseointegration. We expect that our findings will help clinicians and patients choose the best dental treatment and also contribute to further research." This paragraph must to be rewritten. Only report the aim avoiding personal pronoun.
Materials and methods
This section is well described. Nevertheless, please report at the beginning the study design and location were the study was conducted. Later, identify clearly the outcomes measure.
Statistical analysis must to be reported at the end of the material and methods.
Did a sample size calculation performed?
Results
Please check that the results have been reported in the same order in which the outcomes have been presented.
Discussion
The discussion is well written. Please always avoid personal pronoun. Moreover, please add a paragraph with the limitations of the study and further researched aimed to overcome the limitations. Clinical considerations should be reported.
Author Response
Thank you very much for your comments. We have revised our manuscript in accordance with your suggestions as follows:
- Explain QCM in the title and report the study design.
We agreed with your suggestion and changed the title to include the study design
Effects of surface modification on adsorption behavior of cell and protein on titanium surface by using Quartz Crystal Microbalance system
- Early fixation. What does it mean? Primary stability? Please clarify
We agreed with your suggestion and have corrected as follows:
P1 Lines 1
Primary stabillity and osseointegration are major challenges in dental implant treatments
- Report the study design in the title. Authors reported "used to measure the nanogram-level amount of protein and bone marrow cells adhered to the surfaces of Ti implants in real time." Later, authors reported "...QCM sensors (simulating implant materials)" I failed to understand the study design.
We agree with your suggestion and have corrected as follows:
P1 Lines 3-5
In this study, a quartz crystal microbalance (QCM) was used to measure the nanogram-level amount of protein and bone marrow cells adhered to the surfaces of titanium (Ti) surface in real time.
- "surface C content and increase in the O content". Fedine C and O contents
We agree with your suggestion and have corrected our manuscript as follows: 
Page 1 Lines6-8.
The surface-treatment methods resulted in a marked decrease in the surface carbon(C) content and increase in the oxygen (O) content, along with super hydrophilicity. The results of surface analysis, QCM measurements and in vivo showed that the adhesion of both adhesive proteins and bone marrow cells, and the ability to induce hard tissue differentiation, were enhanced after surface treatment.
- "various analyses showed". Which analyses?
We agree with your suggestion and have corrected our manuscript as follows: 
Page 1 Lines 8-10.
The results of QCM measurements showed that adhesion of both adhesive proteins and bone marrow cells was enhanced after surface treatment.
- "We verified..." avoid personal pronoun.
We agree with your suggestion and have corrected our manuscript as follows: 
Page 1 Lines 13-14.
It was verified that QCM is an effective method for analyzing the initial adhesion process on dental implants.
- "Our group..." Avoid personal pronoun
We agree with your suggestion and have corrected our manuscript as follows: 
Page 1 Lines 25-27.
Komasa et al. reported that the surface treatment of titanium and zirconia implant materials by ultraviolet (UV), atmospheric-pressure plasma, and alkali treatments can promote the early formation of hard tissue in the tissue surrounding the implant [6,7].
- Too many references from in vitro studies. Please, also report references from in vivo researches, such as:
Park C-J, Lim JH, Tallarico M, Hwang K-G, Choi H, Cho G-J, et al. Coating of a Sand-Blasted and Acid-Etched Implant Surface with a pH-Buffering Agent after Vacuum-UV Photofunctionalization. Coatings. 2020;10(11):1040.
Tallarico M, Baldini N, Gatti F, Martinolli M, Erta X, Meloni SM, et al. Role of New Hydrophilic Surfaces on Early Success Rate and Implant Stability: 1-Year Post-loading Results of a Multicenter, Split-Mouth, Randomized Controlled Trial. European Journal of Dentistry. 2020 Jul 31;1–10.
We agreed with your suggestion and added it to references 36 and 40.
- "In this study, we compared the initial behavior of bone marrow cells and proteins on the surface of titanium materials treated by UV and atmospheric-pressure plasma treatments, to clarify the differences between these methods. We verified the efficacy of such treatment methods and concluded that atmospheric-pressure plasma treatment provided the best surface properties for promoting early fixation and osseointegration. We expect that our findings will help clinicians and patients choose the best dental treatment and also contribute to further research." This paragraph must to be rewritten. Only report the aim avoiding personal pronoun.
We agree with your suggestion and have corrected our manuscript as follows: 
Page 2 Lines 33-38.
In this study, it was compared that the initial behavior of bone marrow cells and proteins on the surface of titanium materials treated by UV and atmospheric-pressure plasma treatments, to clarify the differences between these methods. It was verified that the efficacy of such treatment methods and concluded that atmospheric-pressure plasma treatment provided the best surface properties for promoting early fixation and osseointegration.
- This section is well described. Nevertheless, please report at the beginning the study design and location were the study was conducted. Later, identify clearly the outcomes measure.
Statistical analysis must to be reported at the end of the material and methods.
Did a sample size calculation performed?
We agreed with your suggestion and have corrected as follows:
2.1.Study design
ALP activity and Calcium deposition of RBM cells on titanium disks subjected to different surface treatments was evaluated. In order to investigate the reason for this difference in biocompatibility, it was focused on the initial behavior of cells and proteins assuming immediately after implantation. By using QCM system, it was compared that adsorption behavior of proteins and RBM cells on Ti-QCM sensors with different surface treatments. In order to examine the results of the QCM analysis, surface analysis and ROS analysis of the material surface were performed, and the events that occurred were considered.
2.11 Statistical Analysis
Each measurement was performed three times and statistical analyses were performed by one-way analysis of variance. When a significant difference was found, Bonferroni's multiple comparison was used. The significance level was < 5%.
11. Please check that the results have been reported in the same order in which the outcomes have been presented.
We agree with your suggestion and have corrected our manuscript as follows:
Page 5Lines 25-26, Page 9Lines 4
The SEM images in Figure 1 show that no significant changes in the surface microstructure of Ti QCM sensors were observed after UV or plasma treatment.
Figure 7 shows the cell morphology observed by phalloidin and DAPI staining after 24 h incubation.
12. The discussion is well written. Please always avoid personal pronoun. Moreover, please add a paragraph with the limitations of the study and further researched aimed to overcome the limitations. Clinical considerations should be reported.
We agree with your suggestion and have corrected our manuscript as follows:
Page 10Lines 16-18,24-27,41-43. Page 11 Lines 1, Page 112 Lines 2-4.
These findings are consistent with those of various previous studies, which report that the hydrophilization of the surface of Ti implants greatly enhances the initial adhesion/proliferation and differentiation-inducing ability of osteoblasts.
Similarly, it was observed that both treatment processes resulted in enhanced ALP activity and calcium precipitation compared to the untreated Ti surface, where the plasma-treated samples showed the best performance.
It was observed that the proteins and cells began to adsorb on the Ti surface immediately, and rapid adsorption was observed for the first 30 min, followed by slower adsorption over the next 30 min.
In this study, it was observed that the morphology of the adsorbed RBM cells on the Ti surfaces.
It was verified that the QCM system was suitable for analyzing the attachment mechanism of bone marrow cells and proteins immediately after implantation.
P12 Lines 34-35
Although the results of this study revealed there was a difference in the effect of each treatment. It was not possible to explain the reason why the amount of adhesion increased rapidly 5 minutes after the appropriate amount. Further consideration is needed on the reason for these results.
Reviewer 2 Report
The manuscript “Effects of surface modification on hard tissue formation on Ti dental implants by using QCM system“ by T. Matsumoto et al. is focused on very attractive research field of dental materials and how to improve an osseointegration process.
The manuscript is well and clearly written and experimental procedures are adequately described. Results of titanium surface modification by two procedures (UV treatment and atmospheric-pressure plasma treatment) can be useful for clinical practice, since these treatments shorten time needed for a formation of stable bone-implant connections.
Remarks:
- The title of the manuscript: The real dental implants were not in the focus of the investigation. A model system (Ti QCM sensor) was used. Therefore, it would be more precise not to use implant in the title and as a key word.
- Page 3, Experimental section: 2.3. Volume of the water drop for contact angle measurements??
- Page 5, Results; 3.2. Chemical composition: The modified Ti surfaces showed significant higher Ti content in comparison to the untreated sample. Please explain.
- Page 6; Results; 3.3. Surface tension: Values of the contact angles for all investigated surfaces??? The data would be useful for readers. Remark: The untreated surface is more hydrophobic compared to treated surfaces. As can be seen from Fig.4. the water contact angle at the untreated surface, q Ë‚ 90Ëš and points to hydrophilic surface.
- Page 6; Results; 3.4. Cell adsorption: In the text it is written: “The adsorbed weight increased rapidly over the first 30 min, followed by a slower increase thereafter up to and beyond 60 min.“ As can be estimated from Fig. 5, the adsorption of almost all cell types increases rapidly over first 5 min or less. Then the increase is slower. Please explain.
Author Response
Thank you very much for your comments. We have revised our manuscript in accordance with your suggestions as follows:
- The title of the manuscript: The real dental implants were not in the focus of the investigation. A model system (Ti QCM sensor) was used. Therefore, it would be more precise not to use implant in the title and as a key word.
We agreed with your suggestion and changed the title and keyword
title
Effects of surface modification on adsorption behavior of cell and protein on titanium surface by using Quartz Crystal Microbalance system
Keyword
titanium UV-treatment, Atmospheric pressure plasma-treatment, QCM measurement
ROS assay
- Page 3, Experimental section: 2.3. Volume of the water drop for contact angle measurements??
We agree with your suggestion and have corrected our manuscript as follows:
Page 2 Lines 28-30
The measurement was performed after 2.6 μL of distilled water was dropped immediately after the surface treatment on the Ti QCM sensors.
- Page 5, Results; 3.2. Chemical composition: The modified Ti surfaces showed significant higher Ti content in comparison to the untreated sample. Please explain.
Thank you for your suggestion. It is considered that carbon stains were deposited on the surface of the pure titanium QCM sensor in the experimental group, and the peak of titanium, which is the backbone, appeared strongly.
- Page 6; Results; 3.3. Surface tension: Values of the contact angles for all investigated surfaces???
We agree with your suggestion and have corrected our manuscript as follows: Page 6 Lines 10-13.
Figure 4 shows the cross-sectional images of water droplets on the three different contact angle of the bare, UV-treated, and plasma-treated Ti QCM sensors of 90.6°, 7.2°, and 0°, respectively. The bare sample was quite hydrophobic, while the surface of the UV-treated Ti was hydrophilic, and that of the plasma-treated Ti was super-hydrophilic (contact angle < 5°).
- Page 6; Results; 3.4. Cell adsorption: In the text it is written: “The adsorbed weight increased rapidly over the first 30 min, followed by a slower increase thereafter up to and beyond 60 min.“ As can be estimated from Fig. 5, the adsorption of almost all cell types increases rapidly over first 5 min or less. Then the increase is slower. Please explain.
We agree with your suggestion and have corrected our manuscript as follows:
P12 Lines 34-35
Although the results of this study revealed there was a difference in the effect of each treatment. It was not possible to explain the reason why the amount of adhesion increased rapidly 5 minutes after the appropriate amount. Further consideration is needed on the reason for these results.